# An Evaluation of Passenger Satisfaction among Users of Huambo Airport in Angola

André Tchoia Relógio [1] and Fernando Oliveira Tavares [2,3,*]

1 Faculty of Economics and Business, University of Vigo, 36310 Vigo, Spain; andre.gaspar.tchoia@uvigo.es
2 Research on Economics, Management and Information Technologies, Department of Economics and Management, Universidade Portucalense, 4200-027 Porto, Portugal
3 Instituto Superior Miguel Torga, Largo da Cruz de Celas nº 1, 3000-132 Coimbra, Portugal
* Correspondence: faotavares@gmail.com or ftavares@upt.pt

**Abstract:** This study aimed to investigate the level of client satisfaction among airline passengers and other users of Huambo's airport in Angola. A quantitative method was used, based on a questionnaire addressed to airline passengers on their trips to Huambo and their use of Huambo's airport. This sample comprises 619 questionnaire answers. As a result of the study, it is possible to relate client satisfaction with the size of the aircraft in question and with the ease of booking a trip. On the contrary, clients become more dissatisfied when the cost of the trip is higher. An analysis of the degree of client satisfaction among airline passengers shows three categories: the waiting time and service at the airline office, the comfort during the trip, and the empathy of the cabin staff. This study is expected to be useful to show the preferences of the clients of this African airport.

**Keywords:** airline passenger satisfaction; airport user satisfaction; Huambo airport; airline companies

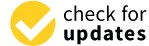



## 1. Introduction

Studies of airports must go beyond the airline traffic they handle. An airport is built to satisfy the demands both of airline companies and passengers [1]. Marketing theory is important for client relations. The management of the airport and the quality of the surrounding market services are important for the clients of all the different activities [1]. The efficiency of an airport is closely related to the number of national and international passengers. Of these two sorts of passengers, ref. [2] considers that international passengers are the most valuable for an airport because they do not have so many other ways of travelling than the airline network.

Air transport has expanded faster than land and water transport [3]. Client satisfaction and client behaviour are conditioned by how the client evaluates the different aspects of the service. Ref. [4] maintains that the degrees of satisfaction and dissatisfaction are interrelated because dissatisfaction can lead to bad physical behaviour on the part of the client and, at the same time, psychic and financial costs for the airport, airline staff, and other clients. Ref. [5] state that little attention has been paid to airport services, which can affect client behaviour. Ref. [6] maintain that passengers normally consider several criteria before taking any action, which confirms the view of [5] on the complexity of their motivation and behaviour.

Ref. [7] see airport services as being composed of two elements: layout and atmosphere. Each element can contribute to the client's evaluation. Refs. [5,8] use the terms "atmospheric" and "servicescape" to describe the service environment. "Atmospheric" refers to the design of the environment, which can evoke specific emotional reactions in the clients and make their continued use more probable [5]. The "servicescape", on the other hand, is the physical and human-made environment in which the goods and services are offered. Ref. [9] maintain that various elements can contribute to satisfaction, such as accessibility, cleanliness, comfort, refreshments, easy parking, clear sign-posting, and

toilet quality. In an airport, ready access to exact information is important. The design and attractiveness of the interior and its furniture also influence the clients' feeling of quality [10].

The service environment does not exist in isolation, but rather is part of the "servicescape" and social environment and normally includes both passengers and staff [11]. According to [12], the interaction between clients can also have an unfavourable influence on satisfaction and relaxation; individual human behaviour is always influenced by the behaviour of others. Airport services can be managed to create and steer the social environment. Bad behaviour among passengers can interfere with the service provided by the staff, which in turn will be reflected in the demand for travelling [13].

The objective of this study is to sample the degree of passenger satisfaction with the airline and their use of Huambo airport in Angola. The study is divided into five sections. After this introduction (1), the relevant literature concerning passenger satisfaction, the management and operation of the airport, and the passengers' opinion of the airport service is reviewed (2). Then, the methodology is presented (3), followed by an analysis and discussion of the results (4). Finally, the conclusions (5) of the study are presented, with a bibliography.

## 2. Survey of Relevant Literature

### 2.1. Client Satisfaction

Passengers who are not satisfied can lead to them behaving badly [5]. Ref. [5] separate the management of client expectations and the client's view of the quality of the service received. This separation is important when defining behaviour in an airport environment. Ref. [5] also observe that the social configurations of the concept of a "servicescape" are not enough to explain dissatisfaction and bad behaviour. These do not arise without a staff mistake.

In Cape Verde, ref. [14] have identified three factors related to client satisfaction with the service provided by the national airline, TACV, Transportadora Aérea de Cabo Verde, comparing the service expected with the most recent experience of this service. Ref. [14] conclude that TACV's client satisfaction depends to a considerable extent on the behaviour of the staff performing the client services. The intangible nature of these services and the influence of the principal competitors are dynamic challenges to TACV. On the one hand, they must consider their passengers' wishes, while on the other hand, as a commercial business, they must meet the demands of the market and the national government.

Ref. [15] consider the regular measurement of the satisfaction of airport users to be a sign of the worries of the airlines, and the airports and their surroundings. To put aside these worries, the air travel business continually studies the efficiency of the relevant airports. As far as check-in and arrivals are concerned, the former is more important, because if this is not carried out according to the programme, it affects the whole voyage, including the flying time [16]. An increase in the time the passengers are on board brings with it expenses for the airline. It is also distressing for the passengers and may upset their travel schedule. Ref. [16] suggest adopting better strategies for the check-in, without ignoring the airline's programme, which is based on the aircraft type, the passengers' profiles and their reasons for travelling, which may be business or amusement. This means a programme that combines different strategies, including the time for the initial check-in and for the relevant aircraft type.

Despite the combined introduction of different strategies to reduce check-in delays and to appreciate passenger satisfaction in airports, it is difficult to discern a common point of view. This is because one of the factors that would make a common point of view possible is the procedures at an airport being a conglomeration of different activities, making it difficult to evaluate the individual components [16]. It is necessary to pay attention to indications, both of quality and quantity, because both of them influence how people appreciate airport service.

The check-in procedure—the reception and registration of passengers—is separate from the security procedure; taken together they give indications of quality and quantity [17]. The standard of service, which the airport offers the passengers, is largely dependent on how the check-in procedure is experienced. Here, client satisfaction is closely related to the influence that the social and physical services have on those who use the airport facilities. Client behaviour is different in each facility, which shows how complex the relations between airport facilities and client satisfaction are. The bad behaviour attributed to passengers often originates from the physical quality of the airport facilities.

The speed of airport services is considered to have improved slower than the speed of land and water transport services [5]. Ref. [18] explain the slow service with boarding and arrival regarding delays in retrieving luggage and frequent and unexplained take-off delays. These can make dissatisfied passengers degenerate into deliberate bad behaviour towards the staff. This is unfortunate because no part of the service offered by the airport exists by itself. The staff's attitudes, enthusiasm, and manner of communicating are transmitted to the passengers [5].

### 2.2. Airport Management

Airport managers have always sought smooth and efficient running [2]. Ref. [18] consider that the management of airports can influence the airline market. The attempts to achieve airport efficiency depend on minimising inputs and maximising outputs. The performance of the organisations currently working at airports can be measured according to signs of the previous efficiency of the relevant airlines [19]. The ability to reduce delays can be different among the airlines and their competitors. Diana emphasises that prompt performance is one of the strategic foundations of any business [20]. Prompt performance impinges upon passenger satisfaction and passenger loyalty.

Apart from the income from airlines, the income of an airport derives from shops and parking. These borderline concessions at the airport can serve as useful criteria. Another significant factor is the physical environment of the airport and its importance in the national airport network, economically and logistically [2]. The efficiency of an airport can also depend on factors that are independent of economic bookkeeping. In the airline business, efficiency can also depend on the extent to which an airport is allowed to manage itself. In this context, an airport may be independent, or it may belong to a group of airports [18].

The opinion of [21] is that investments in the management of an airport and political attention to an airport affect the degree to which air transport meets the challenges of growth and competition. If air transport is to be viable for moving people and goods, punctuality is important, but it also depends on other factors, above all on governmental and private institutions [22]. On this subject, ref. [21] consider that for the success of these institutions, punctuality is a valid indicator of efficiency, but this, in turn, depends on the municipal infrastructure, for example, the access roads and the operational safety of the airline in question [18].

An airline's permanent concern is reducing costs. These costs can have different origins, but the most common are building rents, fuel consumption, maintenance of a fleet of obsolete aircraft, or indeed, the introduction and maintenance of new types of aircraft. All these factors can lead to increased costs for the airline and determine its survival, but they vary from country to country [19].

Ref. [23] consider that the meteorological environment has a large influence on an aircraft's normal performance concerning airports. Ref. [24] point out that airport infrastructure must be monitored frequently to ensure acceptable function. Ref. [25] emphasise that it is necessary to monitor airport management because resources can become obsolete.

Ref. [19] convey that the success of airlines and airport infrastructures is very dependent on operational and economic measures supported by the government. These, in turn, influence the travel market, but it is the number of passengers who pay for a seat on an aircraft, which is fundamental.

Ref. [15] consider that in the airline business admitting new competitors is beneficial for airport users. Ref. [19] point out that both long-distance and short-distance flights must coexist, and this collaboration is valuable because it tends to reduce airline costs.

An airport is a complex administrative system, and as part of the national transport system, it plays an important role in the development and growth of a nation [15,26,27]. Ref. [28] consider that to ensure a good relationship between airport management and the sustainable economic development of a district, there must be a harmonious engagement between the national economic elements, such as the businesses, the families and the other means of transport.

*2.3. Analysis and Synthesis of the Presented Literature*

The management of customer expectations and the perception of service quality can influence the behaviour of travellers in the context of airports [5]. Although the social settings of the airport environment are not sufficient to indicate dissatisfaction and bad behaviour, inadequate staff service can affect customer perception. In the case of TACV, customer satisfaction depends mainly on the characteristics of the service personnel [14].

Airlines and airports should regularly measure user satisfaction [15]. Ref. [16] suggest that the adoption of different strategies can reduce boarding delays, but the perception of quality and passenger satisfaction at airports is still difficult to evaluate. The evaluation of processing capacity in a boarding terminal depends on the level of service offered by the airport to the passenger [17].

Customer satisfaction is directly related to the influence of social and physical services in the airport environment. Customer behaviour at the airport varies in each space, making the relationship between airport services and customer satisfaction complex. Poor behaviour attributed to passengers often originates from the quality of the service provided by the airport system [5].

The service environment can be categorised into physical, social, and ambient elements, and its quality can influence customer satisfaction and behaviour. The physical service environment includes elements such as furniture, decoration, and layout, while the social service environment encompasses the behaviour of fellow travellers and the service team. The ambient service environment includes factors such as noise level, lighting, and music.

Customer satisfaction is influenced by the management of expectations and the perception of service quality. The study by [5] suggests that customer dissatisfaction can lead to negative behaviours, such as theft or aggression, and customer service quality can also influence behaviour. The concept of service quality is defined in [29] and is based on the characteristics of intangibility, inseparability, and heterogeneity. Overall, the text highlights the importance of creating a positive service environment that meets customer expectations and promotes customer satisfaction and positive behaviour.

Airport performance is a constant concern for managers and can affect the market in which airlines operate as they seek to achieve airport efficiency to maximize results [19,20], Diana, 2017. Punctuality is an important factor for passenger satisfaction and loyalty, and can differentiate airlines in the market (Refs. [21,22]. Airport revenues come from two sources: aeronautical and non-aeronautical. The latter originates from the exploitation of shops and parking services. Efficiency can be measured by the degree of autonomy of airports and the relationship between public and private institutions [18]. Cost reduction is also a constant concern for airlines and can be influenced by various factors, such as aircraft obsolescence and the specificities of each country. Weather conditions and infrastructure monitoring also affect airport operations [19]. The success of aviation companies and airport infrastructure depends on operational and economic factors supported by government actions and financial market performance.

## 3. Methodology

After the literature review, to study the degree of satisfaction with the airlines serving Huambo airport, an enquiry form was developed. In addition to questions about respon-

dents' general data, it included specific questions on service quality, as referred to by [29], and airport service quality, as focused by [5]. The form was designed to be completed by hotel guests who had travelled to Huambo by air, and also by people at Huambo airport who had, on some occasion, travelled by air. The airport is a part of Angola's only network for moving people very rapidly. The questionnaire was submitted to a pretest to check for possible flaws that may have occurred, such as inappropriate language or questions that may not relate to the subject under study. After the pretest, the proposed suggestions were taken into consideration, and data were collected by filling out the questionnaires by passengers who used the Huambo Airport. Since it was challenging to collect survey responses only at the Huambo Airport, efforts were made to distribute them at hotels where the airport's passengers stayed.

This form was distributed among hotels in the town of Huambo and at Huambo airport between January and July 2022.

The snowball statistical methodology was used. This technique is also used in quantitative research, which involves identifying participants for the study and obtaining questionnaire responses from them, which will then be processed quantitatively. This technique is useful in research involving specific groups. The process works like a snowball that grows as new participants are identified and added to the study. It should be noted that the use of the snowball methodology may have limitations in terms of sample representativeness, as participants may share common characteristics that are not representative of the general population. Therefore, to avoid this limitation, the sample was stratified by seeking participants in various hotels. Thus, the number of surveys per hotel was Hotel Nino, 68; Hotel Nova Estrela, 68; Guet Hotel, 68; Mess Hotel, 68; Hotel Ritz, 68; Hotel Chimina, 68; and Hotel Ekuikui, 68. In addition, 143 surveys were filled out at Huambo Airport. The enquiry form was completed by 619 persons.

After obtaining the responses, a database was built, and IBM SPSS Statistics 28 software was used for the statistical treatment of the data. Descriptive statistics were used to characterise the respondents and companies and also to analyse the different variables under study.

## 4. Analysis and Review of the Results

*4.1. Descriptive Analysis of Client Satisfaction with Their Use of the Airline and Huambo Airport*

The sample studied consists of 619 persons, aged between 18 and 79 years old, and with a median age of 44 years. Of these 619 persons, 43.6% were men and 56.4% were women. In total, 96.3% were Angolan nationals, and 3.7% were of other nationalities.

Regarding family relationships, 60.7% were married or cohabiting in similar relationships, 21.3% were single, 9.7% were widows or widowers, 5.3% were divorced, and 2.9% had separated. Moreover, in this sample of 619 persons, the average is a five-person family. Regarding education, 54.8% had a first degree, 21.2% had studied up to year 6, 12.9% had a master's or a PhD degree, and 11.1% had year 9. Furthermore, with regard to their household income, 46% considered it satisfactory, 41.4% considered it difficult to manage, 10.5% considered it to be good, and only 2.1% considered it very good.

Concerning their travel companions, 60.3% had travelled with family members, 25.8% travelled alone, 11.6% travelled with business colleagues, 1.9% travelled with friends, and 0.3% could not be classified. In total, 92.4% travelled in the economy class and 7.6% in the luxury class. The distance between home and airport for these 619 persons varied from one to one hundred kilometres, with a median of ten kilometres. In total, 94.7% chose to travel with TAAG, 3.7% with FANA, 1% with SonAir, and 0.6% with other airlines.

Table 1 presents questions related to client satisfaction with the airline in the enquiry form. The highest average scores show that the members of the sample were satisfied or very satisfied. They gave the highest scores to office cleanliness, aircraft comfort, aircraft size and ease of making reasonable reservations. These items have mean values above three on a five-point Likert scale. The five-level Likert scale is widely employed in social sciences to measure individual attitudes and perceptions, providing a clear and quantitative

framework for data collection and analysis. This tool helps researchers gain a deeper understanding of social phenomena and people's opinions. Tranquillity and comfort on board are important factors in the choice of an airline. Less important for client satisfaction with the airline service are affordable trip costs, flight delays, and e-mail service.

**Table 1.** Client satisfaction with the airline.

| | Average | Medium | Mode | Standard Deviation | Not at All Satisfied | Somewhat Satisfied | Undecided | Satisfied | Very Satisfied |
|---|---|---|---|---|---|---|---|---|---|
| Office cleanliness | 3.97 | 4.0 | 4 | 0.882 | 2.6 | 4.7 | 10.7 | 57 | 25.0 |
| Aircraft comfort | 3.87 | 4.0 | 4 | 0.990 | 2.7 | 11.0 | 6.6 | 55.4 | 25.0 |
| Office opening time | 3.86 | 4.0 | 4 | 0.960 | 3.1 | 7.6 | 12.6 | 53.5 | 23.3 |
| Information given | 3.83 | 4.0 | 4 | 0.858 | 2.3 | 6.1 | 14.7 | 60.1 | 16.8 |
| Office appearance | 3.82 | 4.0 | 4 | 0.972 | 3.4 | 8.4 | 12.4 | 54.6 | 21.2 |
| Aircraft size | 3.77 | 4.0 | 4 | 0.911 | 5.2 | 5.7 | 7.4 | 70.3 | 11.5 |
| Service quality | 3.76 | 4.0 | 4 | 0.962 | 2.6 | 11.1 | 12.0 | 56.1 | 18.3 |
| Ground staff politeness | 3.72 | 4.0 | 4 | 0.975 | 5.8 | 7.3 | 9.0 | 65.3 | 12.6 |
| Ground staff efficiency | 3.71 | 4.0 | 4 | 0.887 | 4.2 | 7.4 | 10.7 | 68.8 | 8.9 |
| Cabin staff efficiency | 3.71 | 4.0 | 4 | 0.813 | 1.0 | 6.8 | 25.4 | 53.8 | 13.1 |
| Office staff politeness and helpfulness | 3.70 | 4.0 | 4 | 1.031 | 6.6 | 6.6 | 13.1 | 57.7 | 16.0 |
| Telephone behaviour | 3.68 | 4.0 | 4 | 1.023 | 5.7 | 8.6 | 13.7 | 56.2 | 15.8 |
| Making reservations | 3.64 | 4.0 | 4 | 1.054 | 6.5 | 10.7 | 9.4 | 59.6 | 13.9 |
| Office waiting time | 3.63 | 4.0 | 4 | 1.083 | 5.8 | 12.0 | 13.2 | 51.5 | 17.4 |
| Check-in time | 3.63 | 4.0 | 4 | 1.056 | 6.5 | 10.7 | 10.5 | 58.5 | 13.9 |
| Cabin staff politeness | 3.62 | 4.0 | 4 | 0.885 | 2.4 | 8.7 | 25.0 | 52.5 | 11.3 |
| Embarkation time | 3.60 | 4.0 | 4 | 0.997 | 3.7 | 13.4 | 15.35 | 54.4 | 13.1 |
| E-mail service | 3.33 | 4.0 | 4 | 1.005 | 6.6 | 13.6 | 25.4 | 48.6 | 5.8 |
| Flight delays | 3.14 | 4.0 | 4 | 1.064 | 7.3 | 25.5 | 16.6 | 47.3 | 3.2 |
| Affordable trip cost | 2.48 | 2.0 | 1 | 1.351 | 31.5 | 29.4 | 5.2 | 27.3 | 6.6 |

Source: own elaboration.

Table 2 presents questions related to Huambo airport in the enquiry form. The questions with the highest scores are related to the fact that Huambo has an airport with toilets, electric lighting everywhere, and embarkation areas where it is possible to sit and relax. These items have a mean above three on the Likert scale. Questions with a lower average are related to retail concessions in the waiting areas according to international standards and reasonable access to airport trolleys.

**Table 2.** User evaluation of Huambo airport.

| | Average | Medium | Mode | Standard Deviation | Totally Disagree | Disagree | Undecided | Agree | Totally Agree |
|---|---|---|---|---|---|---|---|---|---|
| Access to clean toilets | 3.77 | 4.0 | 4 | 0.824 | 3.1 | 4.8 | 15.2 | 65.9 | 11.0 |
| Complete illumination | 3.70 | 4.0 | 4 | 0.918 | 3.4 | 9.0 | 13.9 | 61.2 | 12.4 |
| Embarkation seats | 3.67 | 4.0 | 4 | 0.815 | 3.2 | 7.9 | 11.6 | 72.7 | 4.5 |
| Very rapid security | 3.51 | 4.0 | 4 | 1.068 | 6.3 | 12.3 | 18.9 | 49.3 | 13.2 |
| Acceptable access to trolleys | 3.49 | 4.0 | 4 | 1.045 | 4.8 | 11.3 | 29.6 | 38.1 | 16.2 |
| Sufficient waiting room space | 3.35 | 4.0 | 4 | 0.924 | 6.1 | 10.7 | 27.1 | 54.0 | 2.1 |

Source: own elaboration.

Table 3 presents an evaluation of the services provided at Huambo airport. The highest averages correspond to airport service, the airport environment, its cleanliness, and also good fellow-passenger behaviour. The lowest averages are from dissatisfaction with the service provided.

**Table 3.** Evaluation of Huambo airport services.

| | Average | Medium | Mode | Standard Deviation | Totally Disagree | Disagree | Undecided | Agree | Totally Agree |
|---|---|---|---|---|---|---|---|---|---|
| Pleasant airport environment | 3.94 | 4.0 | 4 | 0.771 | 4.5 | 10.7 | 65.4 | 17.9 | 4.5 |
| Clean airport | 3.88 | 4.0 | 4 | 0.746 | 1.3 | 3.4 | 16.3 | 63.7 | 15.3 |
| Travel companions behave in a way I agree with | 3.87 | 4.00 | 4 | 0.783 | 2.1 | 4.2 | 12.6 | 66.6 | 14.5 |
| Fellow passengers behave as I would | 3.81 | 4.0 | 4 | 0.695 | 1.1 | 3.6 | 17.8 | 68.0 | 9.5 |
| Attractive airport environment | 3.80 | 4.0 | 4 | 0.794 | 1.9 | 4.4 | 18.4 | 61.9 | 13.4 |
| Attractive airport exterior | 3.79 | 4.0 | 4 | 0.862 | 2.6 | 7.8 | 11.1 | 65.3 | 13.2 |
| Satisfactory staff treatment by staff | 3.78 | 4.0 | 4 | 0.761 | 2.3 | 3.2 | 18.9 | 65.3 | 1.3 |
| Acceptable noise level | 3.77 | 4.0 | 4 | 0.783 | 1.8 | 5.2 | 18.7 | 63.2 | 11.1 |
| Fellow passengers behaved as hoped | 3.75 | 4.0 | 4 | 0.759 | 1.6 | 6.1 | 16.3 | 67.5 | 8.4 |
| Fully acceptable staff behaviour | 3.75 | 4.0 | 4 | 0.800 | 2.6 | 5.3 | 16.3 | 66.4 | 9.4 |
| Fellow passengers behaved appropriately | 3.74 | 4.0 | 4 | 0.720 | 1.3 | 5.0 | 19.9 | 66.6 | 7.3 |
| Very satisfied with the staff's ability to look after me | 3.73 | 4.0 | 4 | 0.714 | 1.1 | 5.0 | 20.7 | 65.9 | 7.3 |
| Very satisfied with the staff's ability to help me | 3.73 | 4.0 | 4 | 0.770 | 1..3 | 6.8 | 18.6 | 64.3 | 9.0 |
| Attractive airport interior | 3.73 | 4.0 | 4 | 0.838 | 2.1 | 8.9 | 12.9 | 66.1 | 1.0 |
| Fellow passengers behaved politely | 3.71 | 4.0 | 4 | 0.794 | 2.7 | 6.1 | 15.5 | 68.8 | 6.8 |
| Adequate illumination | 3.71 | 4.0 | 4 | 0.832 | 2.1 | 7.4 | 18.6 | 61.41 | 10.5 |
| Generally, the airport looks after its passengers well | 3.68 | 4.0 | 4 | 0.812 | 4.2 | 4.5 | 15.3 | 71.1 | 4.8 |
| Generally, the airport pays a lot of attention to safety | 3.68 | 4.0 | 4 | 1.000 | 6.8 | 5.2 | 14.5 | 59.9 | 13.6 |
| I enjoy being close to other passengers | 3.66 | 4.0 | 4 | 0.805 | 1.6 | 7.4 | 23.9 | 59.1 | 8.9 |
| I have confidence in the staff | 3.64 | 4.0 | 4 | 0.749 | 2.4 | 4.4 | 24.7 | 63.7 | 4.8 |
| Generally, dishonesty is not easy in this airport | 3.64 | 4.0 | 4 | 0.970 | 6.3 | 4.5 | 20.2 | 57.0 | 12.0 |
| I like the airport layout | 3.63 | 4.0 | 4 | 0.989 | 5.5 | 5.7 | 24.6 | 49.3 | 15.0 |
| I like the airport's interior design | 3.60 | 4.0 | 4 | 0.921 | 4.2 | 8.9 | 18.6 | 59.5 | 8.9 |
| The staff seem to be enthusiastic | 3.60 | 4.0 | 4 | 0891 | 4.0 | 6.8 | 22.9 | 57.5 | 8.7 |
| I believe the airport's physical environment is excellent | 3.58 | 4.00 | 4 | 0.927 | 3.7 | 9.5 | 21.8 | 54.9 | 10.0 |
| The airport is not dangerous for its passengers | 3.55 | 4.0 | 4 | 0.948 | 7.1 | 5.2 | 18.8 | 62.8 | 6.0 |
| The music is appropriate | 3.54 | 4.0 | 4 | 0.960 | 6.8 | 5.0 | 23.7 | 56.2 | 8.2 |
| The airport furniture is comfortable | 3.50 | 4.0 | 4 | 0.968 | 4.0 | 13.1 | 20.4 | 53.5 | 9.0 |
| Impressive quality of the physical environment | 3.45 | 4.0 | 4 | 0.986 | 5.5 | 11.3 | 24.2 | 50.6 | 8.4 |
| High standard of the physical environment | 3.45 | 3.0 | 3 | 1.076 | 8.1 | 16.5 | 35.7 | 29.7 | 10.0 |
| I was not very dissatisfied with the airport | 2.86 | 3.0 | 2 | 1.196 | 13.4 | 29.4 | 24.2 | 23.4 | 9.5 |
| I was dissatisfied with the service quality | 2.58 | 2.0 | 2 | 1.158 | 14.9 | 45.9 | 11.6 | 21.5 | 6.1 |
| I was dissatisfied with the level of service | 2.52 | 2.0 | 2 | 1.075 | 15.5 | 44.6 | 15.0 | 22.5 | 2.4 |
| My expectations were not attended to | 2.49 | 2.0 | 2 | 1.030 | 14.4 | 43.6 | 24.4 | 13.9 | 3.7 |

Source: own elaboration.

The passenger behaviour observed by the other passengers is presented in Table 4. Generally the items have low means around one on the Likert scale. The observations "I saw a passenger deliberately steal something at the airport" and "I saw a passenger use violence against airport staff or a passenger" are more frequent than an absence of observations. The answers to the enquiry suggest that the passengers showed no dissatisfaction with the airport service or the airline service.

**Table 4.** Behaviour as observed by passengers.

| | Average | Medium | Mode | Standard Deviation | Never Observed | Very Rarely Observed | Undecided | Sometimes Observed | Often Observed |
|---|---|---|---|---|---|---|---|---|---|
| A passenger receiving unfair preferential treatment did not complain | 2.05 | 2.0 | 1 | 1.024 | 39.1 | 27.1 | 24.7 | 8.1 | 1.0 |
| A passenger was seen making a justified complaint | 1.81 | 2.0 | 1 | 0.987 | 47.5 | 34.7 | 7.6 | 9.2 | 1.0 |
| A passenger was seen taking services without intending to pay for them | 1.53 | 1.0 | 1 | 0.776 | 60.7 | 29.4 | 6.5 | 3.1 | 0.3 |
| A passenger was seen deliberately stealing | 1.46 | 1.0 | 1 | 0.786 | 65.8 | 27.8 | 2.9 | 1.9 | 1.6 |
| A passenger was seen damaging airport property | 1.75 | 1.0 | 1 | 1.041 | 53.6 | 31.5 | 4.4 | 7.6 | 2.9 |
| I have observed travellers vandalizing the airport properties | 1.51 | 1.00 | 1 | 0.833 | 63.7 | 28.1 | 3.2 | 3.7 | 1.3 |
| A passenger was seen using violence against the staff | 1.44 | 1.0 | 1 | 0.731 | 65.3 | 29.6 | 2.3 | 1.81 | 1.1 |

Source: own elaboration.

### 4.2. Factor Analysis of Client Satisfaction with the Airline

A factor analysis presupposes the existence of a smaller number of unobserved variables behind the figures, which together express the sum of these variables. To check that this factor analysis is adequate, we have calculated the Kaiser–Meyer–Olkin (KMO) statistics and applied the Bartlett test. The KMO statistic evaluates the proportion of shared variance among observed variables relative to the total variance of the variables. The closer the KMO value is to 1, the greater the adequacy of the sample for factor analysis or principal component analysis. A value above 0.6 is generally considered acceptable, while values above 0.8 are considered very good. The KMO value is 0.873 and is presented in Table 5, which according to [30,31] allows for a very good factorial analysis. In this case, the Bartlett test value of 0.000 leads to rejecting the hypothesis that a matrix of correlations in the population is the identity matrix, thus showing that the interrelation between some variables is statistically significant. We can therefore conclude that the factor analysis is adequate with regard to satisfaction with the airline. If this were not so, it would be necessary to reconsider this model.

**Table 5.** Overall variations related to passenger satisfaction with the airline.

| | Initial Eigenvalues | | | Additional Uploading Extraction on the Table | | | Additional Uploading Rotation on the Table | | |
|---|---|---|---|---|---|---|---|---|---|
| | Total | % of Variance | % Cumulative | Total | % of Variance | % Cumulative | Total | % of Variance | % Cumulative |
| 1 | 5.691 | 43.778 | 43.778 | 5.691 | 43.778 | 43.778 | 4.176 | 32.125 | 32.125 |
| 2 | 1.648 | 12.680 | 56.459 | 1.648 | 12.680 | 56.459 | 2.482 | 19.092 | 51.217 |
| 3 | 1.061 | 8.162 | 64.620 | 1.061 | 8.162 | 64.620 | 1.742 | 13.403 | 64.620 |

Source: own elaboration.

We can analyse Cronbach's Alpha to confirm that the factors are mutually consistent. Cronbach's Alpha is a statistical measure that assesses the internal consistency of a measurement scale or questionnaire, providing information about the reliability of the obtained results. Cronbach's alpha ranges from 0 to 1, with values closer to 1 indicating higher internal consistency. However, in general, a value above 0.7 is considered satisfactory for most studies. Table 5 allows us to draw out three factors. Table 5 also confirms that the values for two factors are more than 1—the Kaiser criteria. Several attempts were made to make the loading for each variable more than 0.5, as shown in Table 6.

**Table 6.** Matrix of rotative components resulting from customer satisfaction for using air company service.

| Associated Variables | Component | | | Factors Interpretation |
|---|---|---|---|---|
| | 1 | 2 | 3 | |
| E-mail treatment of requests for information | 0.777 | 0.129 | −0.025 | Service and waiting time at airline offices |
| Telephone treatment of requests for information | 0.763 | 0.245 | 0.088 | |
| Office cleanliness | 0.750 | 0.072 | 0.179 | |
| Waiting time at offices | 0.691 | 0.146 | 0.142 | |
| Office opening times | 0.687 | 0.384 | −0.172 | |
| Overall appearance of the offices | 0.684 | 0.441 | 0.048 | |
| Answers to the client's question | 0.669 | 0.163 | 0.130 | |
| Politeness and help towards clients | 0.621 | 0.479 | 0.160 | |
| Ground staff efficiency | 0.277 | 0.793 | 0.105 | Voyage Convenience |
| Aircraft size | 0.161 | 0.765 | 0.105 | |
| Ground staff friendliness | 0.232 | 0.724 | 0.346 | |
| Cabin staff efficiency | 0.133 | 0.146 | 0.861 | Cabin staff friendliness |
| Cabin staff friendliness | 0.055 | 0.147 | 0.855 | |
| Cronbach's alpha | 0.891 | 0.767 | 0.737 | |

Extraction method: analysis of the main component. Rotation method: varimax with normalization of Kaiser converged rotation on table interactions.

Table 5 shows a factor analysis derived from extracting three variables explaining 64.62% of the total variance. The remaining 35.38% not accounted for could be related to other less relevant factors, derived from other combinations of variables.

In Table 6, the three factors are presented and interpreted. Concerning factor 1, an examination of the variables, which together explain this factor, allows for the conclusion that these are associated with the physical aspects and with the reception and waiting time in airline offices. This factor includes the reception of information given by telephone, the cleanliness of the offices, the e-mail treatment of requests for information, the overall appearance of the offices, the waiting time at the offices, the office opening hours, the staff's politeness and helpfulness, and the answers given to requests for information.

The variables included in factor 2 are related to the convenience of the trip: this factor is connected to the efficiency of the ground staff, the aircraft size, and the politeness of the ground staff.

Factor 3 encompasses variables related to the friendliness and efficiency of the cabin staff. The Cronbach's alpha values show that each factor is consistent, as shown in Table 6. The following is a description of how the selected factors were named and interpreted in relation to the principal components.

### 4.3. Factor Analysis of Client Satisfaction with the Airport

Bearing in mind the KMO value of 0.872, which according to [30,31] would permit a final factor analysis and additionally a Bartlett test, which gives a value around 0.000, we can reject the hypothesis that a matrix of correlations in the population is the identity matrix. This shows a statistically significant connection between some of the variables. We can therefore consider the factor analysis adequate with regard to the airport client's satisfaction.

Table 7 allows us to examine five factors. It also confirms that the values attributed to this table are all more than 1—the Kaiser criteria. Several attempts were made to load values greater than 0.5, but they were, in turn, rejected—see also Table 8.

**Table 7.** Total variances relative to Huambo airport.

| | Initial Eigenvalues | | | Additional Uploading Extraction on the Table | | | Additional Uploading Rotation on the Table | | |
|---|---|---|---|---|---|---|---|---|---|
| | Total | % of Variances | Cumulative % | Total | % of Variances | Cumulative % | Total | % of Variances | Cumulative % |
| 1 | 7.625 | 31.770 | 31.770 | 7.625 | 31.770 | 31.770 | 5.041 | 21.005 | 21.005 |
| 2 | 3.006 | 12.527 | 44.297 | 3.006 | 12.527 | 44.297 | 3.348 | 13.949 | 34.954 |
| 3 | 1.868 | 7.784 | 52.081 | 1.868 | 7.784 | 52.081 | 2.918 | 12.160 | 47.114 |
| 4 | 1.713 | 7.136 | 59.217 | 1.713 | 7.136 | 59.217 | 2.120 | 8.835 | 55.949 |
| 5 | 1.275 | 5.312 | 64.529 | 1.275 | 5.312 | 64.529 | 2.059 | 8.581 | 64.529 |

Extraction method: analysis of the main component.

**Table 8.** Rotational component matrix relative to Huambo airport.

| Associated Variables | Component | | | | | Factor Interpretation |
|---|---|---|---|---|---|---|
| | 1 | 2 | 3 | 4 | 5 | |
| The airport's physical environment is excellent | 0.768 | 0.149 | 0.082 | 0.056 | 0.160 | |
| The airport's background music is appropriate | 0.737 | 0.040 | 0.012 | 0.085 | 0.195 | |
| The airport's physical environment is impressive | 0.732 | 0.299 | 0.109 | 0.026 | 0.145 | |
| I like the airport's interior design and furniture | 0.708 | 0.276 | 0.060 | 0.019 | 0.158 | Huambo airport's physical aspect |
| The airport's interior is attractive | 0.704 | 0.189 | 0.029 | 0.096 | 0.26 | |
| The airport's physical environment has a high standard | 0.699 | 0.203 | 0.229 | 0.239 | 0.046 | |
| The airport furniture is comfortable | 0.689 | 0.247 | 0.149 | 0.101 | 0.167 | |
| The airport illumination is adequate | 0.674 | 0.215 | 0.019 | 0.316 | 0.050 | |
| I like the layout of this airport | 0.624 | 0.211 | 0.026 | 0.325 | 0.046 | |
| My fellow passengers behaved appropriately | 0.122 | 0.817 | 0.003 | 0.088 | 0.118 | |
| My fellow passengers behaved as I had hoped | 0.204 | 0.768 | 0.074 | 0.008 | 0.012 | |
| My fellow passengers behaved decently | 0.309 | 0.681 | 0.037 | 0.004 | 0.023 | Fellow passengers |
| My fellow passengers behaved agreeably | 0.231 | 0.634 | 0.143 | 0.258 | 0.176 | |
| My fellow passengers in a way that I like | 0.175 | 0.627 | 0.016 | 0.350 | 0.118 | |
| I like to be near my fellow-passengers | 0.218 | 0.596 | 0.012 | 0–098 | 0.036 | |
| I was not satisfied with the quality of service received | 0.073 | 0.063 | 0.901 | 0.053 | 0.054 | |
| My expectations were not met | 0.018 | 0.039 | 0.900 | 0.036 | 0.050 | Airport satisfaction |
| I was not very dissatisfied with the airport | 0.097 | 0.024 | 0.835 | 0.083 | 0.072 | |
| I was not satisfied with the service received at the airport | 0.057 | 0.026 | 0.672 | 0.094 | 0.011 | |
| Generally, I think it is not easy to be dishonest at this airport | 0.192 | 0.080 | 0.050 | 0.853 | 0.098 | Airport security |
| Generally, I think this airport pays a lot of attention to security | 0.131 | 0.078 | 0.017 | 0.842 | 0.050 | |
| I have confidence in the airport staff | 0.276 | 0.110 | 0.100 | 0.235 | 0.794 | |
| I was very satisfied with the way the staff treated me | 0.236 | 0.130 | 0.080 | 0.227 | 0.788 | Confidence in airport staff |
| I was very satisfied with the staff's ability to meet my needs | 0.280 | 0.104 | 0.044 | 0.240 | 0.718 | |
| Cronbach's alpha | 0.905 | 0.773 | 0.852 | 0.801 | 0.777 | |

Extraction method: analysis of the main component.

The factorial analysis depends on selecting five factors responsible for 64.529% of the total variance—see Table 7. The unexplained variance of 35.471% could be related to other less relevant factors or be the result of other combinations of variables. Cronbach's alpha shows that each factor is consistent—see Table 8.

With regard to factor 1, an examination of the relevant variables shows that they are associated with the physical aspects of the airport. This factor also shows the importance clients attach to the following aspects: a faultless environment, appropriate background music, an appreciation of the interior design and the furniture design, an attractive interior, a high standard in general, comfortable furniture, adequate illumination, and a functional layout.

The variables included in factor 2 are related to the fellow passengers; this factor depends on whether the fellow passengers behaved appropriately, as might be expected, politely, or at least acceptably. They may even behave charmingly and appear to appreciate the company of other passengers.

Concerning factor 3, they are other variables that contribute to client satisfaction with the airport. This factor explains dissatisfaction with the service received or disappointment when the service is not up to the expected level. Factor 4 comprises variables which can contribute to airport safety; the client does not feel in danger of being cheated and feels that the airport is very interested in safety. Finally, with regard to factor 5, this factor is explained by confidence in the staff. The client can be very satisfied with staff treatment.

### 4.4. Statistically Significant Averages for Service Quality, according to Gender

Table 9 presents the difference between statistically significant averages for sundry posts in the enquiry regarding client satisfaction according to gender. The highest averages are derived from women. We can therefore conclude that women are more demanding about service quality at Huambo airport.

**Table 9.** *t*-test for average differences related to service quality, according to gender.

|  | Test of Levane for Variance Equalities | | | | *t*-Test for Average Equalities |
|---|---|---|---|---|---|
|  | *t*-Test | (*p*-Value) | Male | Female | *t*-Test (*p*-Value) |
| Ease of making a reservation within a reasonable time-limit | −4.070 | 0.000 | 3.44 | 3.70 | 0.000 |
| Treatment of information requests by telephone | −3.652 | 0.000 | 3.51 | 3.79 | 0.000 |
| Treatment of information requests by e-mail | −2.127 | 0.000 | 3.24 | 3.41 | 0.034 |
| Office opening times | −4.539 | 0.000 | 3.67 | 4.01 | 0.000 |
| General impressions of the office | −4.899 | 0.000 | 3.60 | 3.98 | 0.000 |
| Politeness and helpfulness at the office | −3.203 | 0.000 | 3.55 | 3.81 | 0.001 |
| Waiting time before check-in | −3.981 | 0.000 | 3.44 | 3.77 | 0.000 |
| Waiting time in the embarkation area | −5.002 | 0.000 | 3.37 | 3.77 | 0.000 |
| Service quality | −3.997 | 0.000 | 3.59 | 3.90 | 0.000 |
| Information is given in response to questions raised | −2.965 | 0.000 | 3.71 | 3.92 | 0.003 |
| Office cleanliness | −3.865 | 0.001 | 3.82 | 4.09 | 0.000 |
| Waiting time at the office | −3.680 | 0.001 | 3.45 | 3.77 | 0.000 |
| Aircraft size | −2.727 | 0.000 | 3.66 | 3.86 | 0.007 |
| Ground staff efficiency | −4.076 | 0.000 | 3.54 | 3.83 | 0.000 |
| Ground staff friendliness | −3.713 | 0.000 | 3.55 | 3.84 | 0.000 |
| Cabin staff efficiency | −3.147 | 0.000 | 3.60 | 3.80 | 0.002 |
| Cabin staff friendliness | −3.340 | 0.000 | 3.48 | 3.72 | 0.001 |
| Flight delays | −4.814 | 0.243 | 2.91 | 3.32 | 0.000 |

Equality of variances/averages. Source: own elaboration.

### 4.5. Statistically Significant Average Differences for Client Satisfaction, according to Travel Class

Table 10 presents the statistically significant average differences for sundry questions in the enquiry related to client satisfaction, according to economy or luxury travel class. The highest averages occur in the luxury class. We can therefore conclude that luxury class passengers are more demanding than economy class passengers, as was previously expected.

**Table 10.** Test-*t* for average for difference in means related to service quality by travel class.

|  | Test of Levane for Variance Equalities | | | | *t*-Test for Average Equalities |
|---|---|---|---|---|---|
|  | *t*-Test | (*p*-Value) | Economy | Luxury | *t*-Test (*p*-Value) |
| Ease of making a reservation within a reasonable time-limit | −2.21 | 0.000 | 3.61 | 3.94 | 0.044 |
| General appearance of the offices | −2.420 | 0.862 | 3.79 | 4.15 | 0.019 |

**Table 10.** *Cont.*

| | Test of Levane for Variance Equalities | | | | *t*-Test for Average Equalities |
| --- | --- | --- | --- | --- | --- |
| | *t*-Test | (*p*-Value) | Economy | Luxury | *t*-Test (*p*-Value) |
| Service quality | −2.400 | 0.016 | 3.74 | 4.09 | 0.017 |
| Information given in response to questions raised | −2.309 | 0.736 | 3.81 | 4.13 | 0.025 |
| Waiting time at the offices | −2.885 | 0.000 | 3.59 | 4.06 | 0.004 |
| Affordable travel costs | −4.933 | 0.343 | 2.41 | 3.32 | 0.000 |
| Aircraft size | −2.989 | 0.678 | 3.74 | 4.12 | 0.004 |
| Ground staff friendliness | −3.730 | 0.057 | 3.68 | 4.13 | 0.000 |
| Cabin staff efficiency | −2.919 | 0.619 | 3.69 | 4.04 | 0.005 |
| Cabin comfort | −2.832 | 0.126 | 3.85 | 4.15 | 0.021 |

Equality of variances/averages. Source: own elaboration.

*4.6. Statistically Significant Average Differences between Genders Regarding the Airport Infrastructures*

Table 11 presents differences in averages statistically relevant for several items of the enquiry on gender differences at the airport of Huambo. As it is observed, there are several items with statistically relevant differences. However, the highest averages in all items are held by females.

**Table 11.** *t*-Test reveals the gender average differences related to Huambo airport.

| | *t*-Test of Levane to Variances Equality | | | | *t*-Test for Averages Equality |
| --- | --- | --- | --- | --- | --- |
| | *t*-Test | (*p*-Value) | Male | Female | *t*-Test (*p*-Value) |
| It is a terminal with clear and available toilets | −2.331 | 0.000 | 3.68 | 3.84 | 0.020 |
| It is totally illuminated | −3.631 | 0.000 | 3.55 | 3.82 | 0.000 |
| It has reasonable seats in the boarding rooms | −2.386 | 0.000 | 3.59 | 3.74 | 0.017 |
| The room occupation process follows the international standard | −4.462 | 0.010 | 3.17 | 3.50 | 0.000 |
| It has a very fast security inspection service | −2.546 | 0.000 | 3.39 | 3.60 | 0.011 |

Equality of variances/averages. Source: own elaboration.

*4.7. Class Average Differences Statistically Significant on the Airport*

Table 12 presents the statistically relevant average differences for diverse items of the enquiry on the airport of Huambo, considering the class within which people travel. As it is observed, there are many items with differences statistically relevant; however, in all the items, the highest averages are held by people who travel in the luxury class.

**Table 12.** Test-*t* for average differences for Huambo airport according to travel class.

| | Test of Levane for Variance Equalities | | | | *t*-Test for Average Equalities |
| --- | --- | --- | --- | --- | --- |
| | *t*-Test (*p*-Value) | (*p*-Value) | Economy | Luxury | *t*-Test (*p*-Value) |
| Clean and accessible toilets | −5.030 | 0.222 | 3.72 | 4.36 | 0.000 |
| Lighting everywhere | −6.661 | 0.015 | 3.63 | 4.53 | 0.000 |
| Embarkation areas with acceptable seating | −2.874 | 0.000 | 3.65 | 4.00 | 0.004 |
| Very rapid inspection and safety check | −3.452 | 0.000 | 3.47 | 3.95 | 0.001 |

Equality of variances/averages. Source: own elaboration.

**5. Conclusions**

This study aimed at investigating the degree of client satisfaction with the airline and with Huambo airport. An enquiry was carried out during the COVID-19 pandemic, which in turn reflects the resurgence of socio-economic activity in Angola. The results of this

enquiry reveal the degree of client satisfaction with the airline TAAG and with the Huambo airport environment. The highest satisfaction with the airline was related to aircraft size and the ease of arranging an acceptably priced trip. The lowest satisfaction was related to the high cost of a trip. When evaluating the airport environment, the highest averages were related to access to toilets, illumination and comfortable waiting areas. The lowest averages were related to the retail concessions in the departure area and to access to airport trolleys.

The average age of the airport's clients was 44 years, of which 56.4% were women. The airline's clients were predominantly, 96.3%, Angolan. In total, 60% of the airline's clients were married or in an equivalent relationship, 61.2% had been clients before the COVID-19 pandemic, 60.3% travelled with their family, 92.4% had chosen the economy class, and 94.7% chose TAAG.

Client satisfaction with the airline has high scores, particularly because the aircraft are spacious and comfortable, and also because booking trips is easy and relatively quick. The physical environment of Huambo airport is also appreciated, with clean toilets, comprehensive illumination, and comfortable seating in the embarkation hall, up to international standards.

An explorative factor analysis was carried out, to measure client satisfaction with the airline in Huambo. Three factors account for 64.62% of the variance in the choice of airline. The first factor is the physical aspects of the airline offices, the reception and the waiting time. To this can be added office cleanliness, the information given over the telephone, and information provided by e-mail, which can reduce the time spent at the offices, as well as the politeness and helpfulness there.

The second factor is related to the trip's convenience—ground staff efficiency and aircraft size. The third factor is the empathy and efficiency of the cabin staff. The enquiry also indicates client reactions according to gender. Women gave the highest scores, which means that they valued these issues more than men did. These issues were ease of making a reservation, treatment of requests for information by e-mail and by telephone, office opening hours, friendly and helpful service, and waiting time at check-in and embarkation.

Similarly, the statistically significant average differences in client reactions were analysed according to the economy and luxury classes. Passengers in the luxury class are interested in the general appearance of the offices, the service quality, the aircraft size and the comfort of the aircraft. Passengers in economy class are interested in what they had heard about the appearance of the offices, the service quality, and the size and comfort of the aircraft. We can conclude that the luxury class passengers can well afford the cost of the trip and are more demanding of service quality.

Customer satisfaction in the airport environment is influenced by several factors, including the quality of services provided by staff, management of customer expectations, and perceived service quality. These factors can impact customer behaviour, and creating a positive service environment is important to promote customer satisfaction and positive behaviour. Efficient airport management is essential for the aviation industry, and efficiency in both aeronautical and non-aeronautical revenue is important. Airport management can impact the airline market, and punctuality is an important factor for passenger satisfaction. Additionally, investments in infrastructure, monitoring, and organizational and community development are important for the success of airlines and airport infrastructure.

As theoretical and practical implications of this study, it is clear that the importance of service quality must be emphasized, seeking continuous improvement throughout the air transport environment. Thus, the role of the State, namely in improving airport infrastructure, is essential. In these improvements, special attention should be given to enhancements in boarding areas and furnishings in waiting rooms, access to baggage trolleys, as well as interior and exterior lighting at the airport; however, airlines must also frequently measure the quality of their services, as only then can they achieve high customer loyalty. Another concern resulting from the study, and which will be important to address, especially in a growing country, is the price of air travel. Despite being a fast and safe mode of transportation, its demand is not growing more due to its high price.

　　　　A restriction for this study was that it had to be carried out during the COVID-19 pandemic. It would therefore be valuable to adjust the enquiry and repeat it outside the influence of the pandemic. As a future work, we understand that the positive and negative externalities of the impact of the Huambo airport's location on the lives of city residents should be studied. Another study of interest to be conducted will be on the economic and social impact of the Huambo airport in this province of Angola.

**Author Contributions:** Conceptualization, A.T.R. and F.O.T.; methodology, A.T.R. and F.O.T.; software A.T.R. and F.O.T.; validation, A.T.R. and F.O.T.; formal analysis, A.T.R. and F.O.T.; investigation, A.T.R. and F.O.T.; data curation, A.T.R.; writing—original draft preparation, A.T.R.; writing—review and editing, A.T.R. and F.O.T.; visualization, A.T.R. and F.O.T.; supervision, F.O.T. All authors have read and agreed to the published version of the manuscript.

**Funding:** This research received no external funding.

**Data Availability Statement:** Not applicable.

**Conflicts of Interest:** The authors declare no conflict of interest.

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
