# Peer review of "An Evaluation of Passenger Satisfaction among Users of Huambo Airport in Angola"

_urbansci, doi:10.3390/urbansci7020057_

Round 1
Author Response
Dear Reviewer,
Thank you very much for your suggestions and comments. Attached you may find a file with our responses.
We have covered all your suggestions and hope our responses meet your standards. We are, of course, at your disposal for any further changes you believe are necessary.
Best regards.

Reviewer 2 Report
The subject of customer satisfaction, regardless of the area to which it refers, are relevant, current, and interesting topic both for the academic community and for the business world, as it is an area that allows the customer's perception of the value that the good or service offers you a crucial area for profit and business success.
The work is well structured. The abstract, introduction, and literature review are written in a thoughtful and balanced manner.
The abstract presents the objective, the theoretical framework, the methodology, and the main conclusions. The authors could have been clearer in listing the main contributions or social impacts.
The introduction is well structured and fulfills its objective, with the presentation of the theme, objectives, and structuring of the work. The methodology is correct; the authors carry out a quantitative study in the analysis of responses to a questionnaire, having obtained 619 responses, which I consider a good number of responses. The authors initially carry out a descriptive analysis and then a factor analysis, presenting the results with comments, in a parsimonious way.
The conclusion is presented in an objective and parsimonious way, without being long and tedious, the authors refer to the main conclusions and point out limitations and future paths, I believe however that at this point the authors could have gone further, pointing out future investigations beyond repetition out of the pandemic. The bibliography is current; more than 43% are from the last 5 years, which confirms the current importance of the topic.
Author Response

(The authors gave the same response as above.)

Author Response

(The authors gave the same response as above.)

Reviewer 4 Report
This paper surveyed 619 users of Angola's Huambo Airport on the customer satisfaction of airline passengers. However, there are some areas that need to be strengthened. Please find out the attached file.

Minor editing of the English language required
Author Response

(The authors gave the same response as above.)

Reviewer 5 Report
TITLE: An Evaluation of Passenger-Satisfaction among Users of 2 Huambo Airport in Angola
Reviewer coment:
- This paper correspond for scope of journal.
- The title corresponds to the content of the paper.
- This study represents significant contribution for the development and improving airport’s service and satisfaction of airline passengers.
- The main question of paper addressed to study degree of passenger satisfaction with the airline and their use of Huambo airport in Angola which can contribute to development of economy in Angola.
- The aim of research is clearly and fully pointed out on the end of chapter of Introduction
- The chapter Material and methods contain wrong text!
- Key words are appropriate.
- Scientific methodology is applied correctly for this type of study.
- Results are clearly presented and discussed.
- Tables, figures, pictures are clear.
- Conclusions are derived on the basis of research results.
- This study represents complementary to the previous ones. +
- Manuscript is acceptable after minor corrections.
Sugestion:
In line 62 – 67 - should be delete text: “The study is divided into five sections After this introduction (1), the relevant literature concerning passenger satisfaction, the management and operation of the airport, and the passengers’ opinion of the airport service is reviewed (2). Then the methodology is presented (3), followed by an analysis and discussion of the results (4). Finally, the conclusions (5) of the study are presented, with a bibliography.”
In line 69 – 83 - should be delete because this text contains instruction how to write Material and methosd as well as some suggestion!
Author Response

(The authors gave the same response as above.)

Round 2
Reviewer 4 Report
It was revised and supplemented by reflecting the request for modification and accepted in its present form.
It is fine with the Quality of the English Language.